# Lung function and peak oxygen uptake in chronic obstructive pulmonary disease phenotypes with and without emphysema

Øystein Rasch-Halvorsen[1,2]*, Erlend Hassel[3], Ben M. Brumpton[2,4,5], Haldor Jenssen[6], Martijn A. Spruit[7,8,9], Arnulf Langhammer[10], Sigurd Steinshamn[1,2]

1 Department of Circulation and Medical Imaging, Faculty of Medicine and Health Sciences, NTNU, Norwegian University of Science and Technology, Trondheim, Norway, 2 Clinic of Thoracic and Occupational Medicine, St. Olavs Hospital, Trondheim University Hospital, Trondheim, Norway, 3 Norwegian Armed Forces Occupational Health Service, Trondheim, Norway, 4 K.G. Jebsen Center for Genetic Epidemiology, Department of Public Health and Nursing, Faculty of Medicine and Health Sciences, NTNU, Norwegian University of Science and Technology, Trondheim, Norway, 5 MRC Integrative Epidemiology Unit, School of Social and Community Medicine, University of Bristol, Bristol, United Kingdom, 6 Telemark Heart Lung and Blood Institute, Skien, Norway, 7 Department of Research and Development, CIRO, Horn, The Netherlands, 8 Department of Respiratory Medicine, Maastricht University Medical Centre, NUTRIM School of Nutrition and Translational Research in Metabolism, Maastricht, The Netherlands, 9 REVAL–Rehabilitation Research Center, BIOMED–Biomedical Research Institute, Faculty of Rehabilitation Sciences, Hasselt University, Diepenbeek, Belgium, 10 Department of Public Health and Nursing, Faculty of Medicine and Health Sciences, NTNU, Norwegian University of Science and Technology, Trondheim, Norway

☯ These authors contributed equally to this work.
* oystein.rasch-halvorsen@ntnu.no

**Data Availability Statement:** All relevant data are within the manuscript and its Supporting Information files.

## Abstract

Previous studies of associations of forced expiratory lung volume in one second ($FEV_1$) with peak oxygen uptake ($VO_{2peak}$) in chronic obstructive pulmonary disease (COPD) have not taken sex, age and height related variance of dynamic lung volumes into account. Nor have such demographic spread of spirometric measures been considered in studies comparing $VO_{2peak}$ between COPD phenotypes characterized by degree of emphysema. We aimed to assess the association of $FEV_{1Z\text{-}score}$ with $VO_{2peak}$ in COPD (n = 186) and investigate whether this association differs between emphysema (E-COPD) and non-emphysema (NE-COPD) phenotypes. Corresponding assessments using standardized percent predicted $FEV_1$ ($ppFEV_1$) were performed for comparison. Additionally, phenotype related differences in $VO_{2peak}$ were compared using $FEV_{1Z\text{-}score}$ and $ppFEV_1$ as alternative expressions of $FEV_1$. E-COPD and NE-COPD were defined by transfer factor of the lung for carbon monoxide below and above lower limits of normal (LLN), respectively. The associations were assessed in linear regression models. One unit reduction in $FEV_{1Z\text{-}score}$ was associated with 1.9 (95% CI 1.4, 2.5) ml/kg/min lower $VO_{2peak}$. In stratified analyses, corresponding estimates were 2.2 (95% CI 1.4, 2.9) and 1.2 (95% CI 0.2, 2.2) ml/kg/min lower $VO_{2peak}$ in E-COPD and NE-COPD, respectively. The association did not differ statistically by COPD phenotype (p-value for interaction = 0.153). Similar estimates were obtained in analyses using standardized $ppFEV_1$. Compared to NE-COPD, $VO_{2peak}$ was 2.2 (95% CI 0.8, 3.6) and 2.1 (95% CI 0.8, 3.5) ml/kg/min lower in E-COPD when adjusted for $FEV_{1Z\text{-}score}$ and $ppFEV_1$, respectively. In COPD, $FEV_{1Z\text{-}score}$ is positively associated with $VO_{2peak}$. This association

**Funding:** ØR-H received a research grant (Grant 2015/FO5150) from the Dam Foundation (https://www.dam.no/logo/). The funders had no role in study design, data collection and analysis, decision to publish, or preparation of the manuscript.

**Competing interests:** The authors have declared that no competing interests exist.

was stronger in E-COPD but did not differ statistically by phenotype. Both the association of $FEV_1$ with $VO_{2peak}$ and the difference in $VO_{2peak}$ comparing COPD phenotypes seems independent of sex, age and height related variance in $FEV_1$. Mechanisms leading to reduction in $FEV_1$ may contribute to lower $VO_{2peak}$ in E-COPD.

## Introduction

In current strategic documents provided by the Global Initiative for Chronic Obstructive Lung Disease (GOLD), the post-bronchodilator ratio of forced expiratory lung volume in one second/forced vital capacity ($FEV_1$/FVC) is the key diagnostic variable defining chronic obstructive pulmonary disease (COPD) [1]. Reduced $FEV_1$ denotes the severity of airway obstruction and may reflect the pathophysiological mechanism of expiratory flow limitation (EFL) at rest or during exercise, characteristic of this disease [2].

Dynamic hyperinflation (DH), inflicting restrictive constraints on tidal volume expansion, is a functional consequence of EFL [3] and associated with dyspnea and exercise limitation in COPD [4]. Exercise intolerance from dyspnea on exertion is a major manifestation of COPD leading to physical inactivity, deconditioning and reduced exercise capacity [5]. Assessed by cardiopulmonary exercise testing (CPET), peak oxygen uptake ($VO_{2peak}$) is a direct measure of exercise capacity and reduced values are associated with increased mortality in COPD [6].

Although integrated in the concept of ventilatory limitation to exercise, variables of airway obstruction and exercise capacity are not strongly associated in COPD [7–9]. However, previous studies have not adequately taken sex, age and height related variance of dynamic lung volumes into account when reporting associations of $FEV_1$ with $VO_{2peak}$ among those with COPD. Furthermore, the specificity of the diagnostic criterion of airway obstruction, when defined by the fixed ratio of post-bronchodilator $FEV_1$/FVC less than 0.70, is age dependent and may lead to over-diagnosis of COPD in an older population [10–12]. In contrast to percent predicted $FEV_1$ (pp$FEV_1$), and the fixed ratio of $FEV_1$/FVC, Z-scores of these measures provide spirometric expressions that are comparable independent of sex, age and height. Reference equations for the calculation of Z-scores of dynamic lung volumes are readily available [13], and both the American Thoracic Society (ATS) and the European Respiratory Society (ERS) have endorsed defining the lower limit of normal (LLN) of lung function measures by the fifth percentile (z-score = -1.645) [14].

COPD is recognized as a heterogeneous disease and different phenotypes have been proposed [15]. In patients with predominantly emphysema (E-COPD), destruction of the pulmonary parenchyma leads to increased compliance from loss of elastic recoil pressure predisposing dynamic compression of the airways. The non-emphysema (NE-COPD) phenotype is characterized by airway inflammation with intrinsic caliber reduction and increased airflow resistance. These two phenotypes represent pathophysiological diversity in COPD highlighting mechanisms contributing to reduced $FEV_1$ and expiratory flow limitation.

The transfer factor of the lung for carbon monoxide ($T_LCO$) has been used to differentiate E-COPD from NE-COPD [14]. Patients with reduced $T_LCO$ have been shown to have lower $VO_{2peak}$ than patients with more preserved $T_LCO$ when compared at similar pp$FEV_1$ [4]. It is unknown whether this difference in $VO_{2peak}$ is related to normalization of $FEV_1$ by pp$FEV_1$, and/or whether the association of $FEV_1$ with $VO_{2peak}$ is stronger in E-COPD than in NE-COPD. Alternatively, mechanisms unrelated to $FEV_1$ may explain the difference in exercise capacity comparing these COPD phenotypes.

Classification of patients by COPD phenotypes should be related to differences in clinically important outcomes [16]. New knowledge on associations between major diagnostic and prognostic measures in readily identifiable phenotypes of this multifaceted disease is likely to be of scientific and clinical value. We aimed to assess the association of $FEV_{1Z\text{-score}}$ with $VO_{2peak}$ in COPD and investigate whether this association differs comparing phenotypes with and without emphysema. Additionally, we explored the potential influence of sex, age and height related variance in $FEV_1$ on both the association of $FEV_1$ with $VO_{2peak}$ and on the difference in $VO_{2peak}$ between E-COPD and NE-COPD.

## Materials and methods

### Study population

We assessed patients referred for evaluation with pulmonary function tests (PFT) and CPET due to exercise intolerance at a private specialist clinic in Norway (Telemark Heart Lung and Blood Institute) between May 1999 and November 2014. The clinical database was accessed in the time period between the fifteenth of July and the thirtieth of August 2015. Patients with COPD (n = 301) were included.

In this study, the diagnostic criterion of airway obstruction in COPD was defined by the post-bronchodilator $FEV_1/FVC$ less than LLN. E-COPD and NE-COPD phenotypes were defined by $T_LCO$ below and above LLN, respectively [14]. LLN was defined as the fifth percentile (z-score = -1.645). Predicted values and Z-scores of dynamic lung volumes and lung diffusing capacity measures were calculated using the Global Lung Function Initiative (GLI) software [17, 18].

Patients > 80 years of age (n = 6) were excluded due to having an age above the valid range of the GLI 2017 reference values for $T_LCO$. Patients ≤ 45 years of age (n = 28) were excluded due to low prevalence of COPD in this age group. We excluded 33 patients with a clinical diagnosis of asthma. In order to ensure uniform modality of exercise testing, 13 patients were excluded due to having performed the CPET on a treadmill. Furthermore, 19 patients were excluded because of termination of CPET due to chest pain and electrocardiogram (ECG) changes suggestive of myocardial ischemia (n = 2), arrhythmia or high blood pressure (n = 9), leg/knee or hip-pain (n = 4), caution due to known aortic valve disease or aortic aneurysm (n = 2), inadequate cycling technique (n = 1) or lack of motivation (n = 1). We also excluded 11 patients who performed PFT and CPET more than 3 months apart.

Three patients were excluded due to missing values on $T_LCO$ and two patients due to missing values on smoking status. After exclusions, 186 patients remained in the statistical analyses.

### Pulmonary function tests

Spirometry and lung diffusing capacity (Vmax Legacy/Spectra 229; SensorMedics) were tested on separate visits, but no more than 3 months prior to CPET (exclusion criterion). The procedures adhered to recommendations provided by the American Thoracic Society/European Respiratory Society guidelines [19, 20].

### Cardiopulmonary exercise test

An incremental exercise test was performed on a cycle ergometer (ER900; Ergoline). The work rate was increased by 5–25 W/minute on an individualized basis aiming for test termination after 8–12 minutes. The patients reported the dominating symptom limiting further exercise as dyspnea, leg discomfort/fatigue or other. The reasons for test termination in the latter category included pain in lower extremities (n = 9), vertigo (n = 4) and attainment of estimated maximal heart rate (n = 1).

Breath by breath measurements (Vmax Legacy/Spectra 229; SensorMedics) of oxygen uptake ($VO_2$) and carbon dioxide output ($VCO_2$) were averaged over 20 seconds. The highest value of $VO_2$ was normalized by bodyweight and termed $VO_{2peak}$.

Pulse oximetry ($S_pO_2$) was measured continuously from rest to test termination using a finger probe (Model 340; Palco Labs/8600; Nonin). The minimum value was termed $S_pO_{2min}$.

Maximal voluntary ventilation (MVV) was estimated by $FEV_1$ x 40 [21]. Ventilatory reserve (VR) was calculated as VR = 1 –Minute ventilation at peak exercise ($VE_{peak}$)/MVV [22]. Maximal heart rate ($HR_{max}$) was estimated by 220 –age. Heart rate reserve (HRR) was calculated as HRR = 1 –Heart rate at peak exercise ($HR_{peak}$)/$HR_{max}$ [22].

## Statistical analyses

Descriptive statistics, calculated as mean and standard deviation (SD) or number of observations and percentages, are reported for both the total sample and the sample stratified by COPD phenotype. Lung function and exercise variables were compared between patients with E-COPD and NE-COPD using independent samples t-tests.

The association of $FEV_{1Z-score}$ with $VO_{2peak}$ was investigated in linear regression models. In a total sample model, $VO_{2peak}$ was regressed on $FEV_{1Z-score}$ and COPD phenotype. Body mass index (BMI), smoking status (former or current), beta-blocker and bronchodilator treatment (both dichotomous), were considered potential confounders. In separate models adjustments were also made for sex and age due to known strong associations with $VO_{2peak}$. An interaction term ($FEV_{1Z-score}$ x COPD phenotype) was included in a separate analysis to investigate effect modification of COPD phenotype on the association of $FEV_{1Z-score}$ with $VO_{2peak}$. Due to potential lack of power in the interaction analysis, the associations of $FEV_{1Z-score}$ with $VO_{2peak}$ were also estimated in analyses stratified by COPD phenotype. Corresponding analyses with standardized $ppFEV_1$, calculated as (observed value–sample mean)/SD, were performed for comparison. Regression coefficients (β) with 95% confidence intervals (CI) are reported.

Linear regression models were also used to investigate whether the difference in $VO_{2peak}$ between E-COPD and NE-COPD is influenced by sex, age and height related variance in $FEV_1$. $VO_{2peak}$ was regressed on COPD phenotype, $FEV_{1Z-score}$ and $ppFEV_1$ in separate models adjusted for sex, age, BMI, smoking status, beta-blocker and bronchodilator treatment.

The potential influence of effort dependent underestimation of maximal exercise capacity from $VO_{2peak}$ was evaluated in a sensitivity analysis including only those with reduced VR and/or HRR. $VO_{2peak}$ was regressed on $FEV_{1Z-score}$ and COPD phenotype with adjustment for age, sex, BMI, smoking status, beta-blocker and bronchodilator treatment.

Residual plots were inspected in all models, and no violations of the assumptions of linear regression were uncovered. Statistical analyses were performed with IBM SPSS Statistics Version 25 (IBM Corp., Armonk, NY, USA).

## Ethics approval

The Regional Committee for Medical and Health Research Ethics approved this retrospective study and the use of anonymous data without informed consent (REC-Central 2012/673).

## Results

### Clinical characteristics and PFT measures

The majority of patients were men, in both the total cohort (65%) and in subgroups stratified by COPD phenotype (63% vs. 67% in E-COPD and NE-COPD, respectively). Mean values of age, BMI and $SpO_2$ at rest were similar in both phenotypes (63.9 vs. 60.4 years, 25.6 vs. 26.7

kg/m$^2$ and 95.1 vs. 96.0% in E-COPD and NE-COPD, respectively). All patients were former or current smokers with similar proportions in both phenotypes (current smokers; 51% vs. 49% in E-COPD and NE-COPD, respectively). The majority of patients did not receive any beta-blocker (67% vs. 76% in E-COPD and NE-COPD, respectively), but had ongoing bronchodilator treatment (73% vs. 53% in E-COPD and NE-COPD, respectively) (Table 1).

Compared to patients with NE-COPD, patients with E-COPD had lower $FEV_{1Z\text{-score}}$ (difference 0.69, 95% CI 0.36, 1.02, p < 0.001), lower $FVC_{Z\text{-score}}$ (difference 0.42, 95% CI 0.05, 0.79, p = 0.028), lower $(FEV_1/FVC)_{Z\text{-score}}$ (difference 0.62, 95% CI 0.38, 0.86, p < 0.001), lower $KCO_{Z\text{-score}}$ (difference 1.74, 95% CI 1.40, 2.07, p < 0.001), lower $VA_{Z\text{-score}}$ (difference 0.91, 95% CI 0.56, 1.27, p < 0.001) and, by definition, lower $T_LCO_{Z\text{-score}}$ (difference 2.41, 95% CI 2.14, 2.68, p < 0.001) (Table 1).

## CPET measures

The majority of patients reported test termination due to dyspnea in both phenotypes, but the proportion was higher in E-COPD than in NE-COPD (67 vs. 52%). The proportion reporting test termination due to leg fatigue was lower in E-COPD than in NE-COPD (26 vs. 40%) (Table 2).

**Table 1. Clinical characteristics and pulmonary function test measures in total cohort and stratified by emphysema (E) and non-emphysema (NE) COPD phenotypes.**

|  | Total | E-COPD | NE-COPD | p-value |
|---|---|---|---|---|
|  | (n = 186) | (n = 101) | (n = 85) |  |
| Men/women | 121 (65)/65 (35) | 64 (63)/37 (37) | 57 (67)/28 (33) |  |
| Age (years) | 62.3 ± 8.9 | 63.9 ± 9.3 | 60.4 ± 8.0 |  |
| BMI (kg/m$^2$) | 26.1 ± 3.6 | 25.6 ± 3.7 | 26.7 ± 3.4 |  |
| SpO$_2$ (%)[a] | 95.5 ± 2.0 | 95.1 ± 2.2 | 96.0 ± 1.6 |  |
| Former/current smokers | 92 (49)/94 (51) | 49 (49)/52 (51) | 43 (51)/42 (49) |  |
| Beta-blocker | 53 (28) | 33 (33) | 20 (24) |  |
| Bronchodilator | 119 (64) | 74 (73) | 45 (53) |  |
| FEV$_1$ (L) | 2.06 ± 0.77 | 1.82 ± 0.73 | 2.33 ± 0.73 |  |
| FEV$_{1Z\text{-score}}$ | -2.16 ± 1.21 | -2.48 ± 1.24 | -1.78 ± 1.06 | < 0.001 |
| ppFEV$_1$ (%) | 65.8 ± 19.8 | 59.8 ± 20.7 | 72.8 ± 16.2 |  |
| FVC (L) | 3.77 ± 1.10 | 3.58 ± 1.02 | 4.00 ± 1.15 |  |
| ppFVC (%) | 93.4 ± 19.4 | 90.5 ± 20.2 | 96.8 ± 17.8 |  |
| FVC$_{Z\text{-score}}$ | -0.45 ± 1.29 | -0.65 ± 1.33 | -0.23 ± 1.21 | 0.028 |
| FEV$_1$/FVC | 0.54 ± 0.10 | 0.50 ± 0.11 | 0.58 ± 0.07 |  |
| (FEV$_1$/FVC)$_{Z\text{-score}}$ | -2.84 ± 0.92 | -3.12 ± 1.02 | -2.50 ± 0.63 | < 0.001 |
| KCO (mmol/min/kPa/L) | 6.07 ± 2.05 | 4.80 ± 1.20 | 7.57 ± 1.82 |  |
| KCO$_{Z\text{-score}}$ | -1.98 ± 1.54 | -3.08 ± 1.10 | -0.67 ± 0.77 | < 0.001 |
| VA (L) | 5.44 ± 1.33 | 5.13 ± 1.22 | 5.82 ± 1.38 |  |
| VA$_{Z\text{-score}}$ | -0.70 ± 1.30 | -1.12 ± 1.29 | -0.21 ± 1.14 | < 0.001 |
| T$_L$CO (mmol/min/kPa) | 1.13 ± 0.29 | 0.96 ± 0.23 | 1.32 ± 0.23 |  |
| T$_L$CO$_{Z\text{-score}}$ | -1.47 ± 1.43 | -2.27 ± 1.22 | -0.53 ± 1.05 | < 0.001 |

Values are mean ± standard deviation or number of observations (percentages). COPD–chronic obstructive pulmonary disease, BMI–body mass index, SpO$_2$ –pulse oximetry, FEV$_1$ –forced expiratory lung volume in one second, pp–percent predicted, FVC–forced vital capacity, KCO–diffusion constant for carbon monoxide, VA– alveolar volume, T$_L$CO–transfer factor of the lung for carbon monoxide.

[a] n = 181: Emphysema/non-emphysema = 98/83.

**Table 2. Cardiopulmonary exercise test measures in total cohort and stratified by emphysema (E) and non-emphysema (NE) COPD phenotypes.**

| | | Total | E-COPD | NE-COPD | p-value |
|---|---|---|---|---|---|
| | | (n = 186) | (n = 101) | (n = 85) | |
| Test termination | | | | | |
| | Dyspnea | 112 (60) | 68 (67) | 44 (52) | |
| | Leg fatigue | 60 (32) | 26 (26) | 34 (40) | |
| | Other[a] | 14 (8) | 7 (7) | 7 (8) | |
| $VO_{2peak}$ (ml/kg/min) | | 20.9 ± 6.5 | 18.7 ± 5.5 | 23.5 ± 6.6 | < 0.001 |
| $RER_{peak}$ | | 1.11 ± 0.13 | 1.09 ± 0.14 | 1.13 ± 0.13 | 0.031 |
| HRR (%)[b] | | 9.9 ± 12.7 | 12.7 ± 12.9 | 6.6 ± 11.6 | 0.001 |
| $VE_{peak}$ (L/min) | | 67.9 ± 25.2 | 61.1 ± 22.6 | 76.0 ± 25.9 | < 0.001 |
| MVV (L/min) | | 82.2 ± 30.8 | 72.9 ± 29.1 | 93.3 ± 29.2 | |
| VR (%) | | 14.5 ± 20.9 | 11.9 ± 23.0 | 17.5 ± 17.7 | 0.073 |
| $SpO_{2min}$ (%)[c] | | 94.3 ± 3.2 | 93.5 ± 3.6 | 95.3 ± 2.2 | < 0.001 |

Values are mean ± standard deviation or number of observations (percentages). COPD–chronic obstructive pulmonary disease, $VO_{2peak}$–oxygen uptake at peak exercise, $RER_{peak}$–respiratory exchange ratio at peak exercise, HRR–heart rate reserve, $VE_{peak}$–minute ventilation at peak exercise, MVV–maximal voluntary ventilation, VR–ventilatory reserve, $SpO_{2min}$–minimum value pulse oximetry.

[a]Pain in lower extremities (n = 9). Vertigo (n = 4). Attainment of estimated maximal heart rate (n = 1).

[b]n = 185: Emphysema/non-emphysema = 100/85.

[c]n = 185: Emphysema/non-emphysema = 101/84.

Compared to patients with NE-COPD, patients with E-COPD had lower $VO_{2peak}$ (difference 4.8 ml/kg/min, 95% CI 3.0, 6.5, p < 0.001), lower $RER_{peak}$ (difference 0.04 95% CI 0.004, 0.08, p = 0.031), higher HRR (difference 6.1%, 95% CI 2.5, 9.6, p = 0.001), lower $VE_{peak}$ (difference 14.8 l/min, 95% CI 7.8, 21.9, p < 0.001), lower VR (difference 5.5%, 95% CI -0.5, 11.5, p = 0.073) and lower $S_pO_{2min}$ (difference 1.7%, 95% CI 0.9, 2.6, p < 0.001) (Table 2).

## Associations of $FEV_{1Z-score}$ with $VO_{2peak}$ and effect modification by COPD phenotype

Adjusted for COPD phenotype, sex, age, BMI, smoking status, beta-blocker and bronchodilator treatment, one unit reduction in $FEV_{1Z-score}$ (i.e. one SD reduction in $FEV_1$) was, on average, associated with 1.9 (95% CI 1.4, 2.5) ml/kg/min lower $VO_{2peak}$ (Table 3).

**Table 3. Associations of $FEV_{1Z-score}$ with $VO_{2peak}$ (ml/kg/min) in total cohort and stratified by emphysema (E) and non-emphysema (NE) COPD phenotypes.**

| | | Model 1[a] | | Model 2[b] | | Model 3[c] | | Model 4[d] | |
|---|---|---|---|---|---|---|---|---|---|
| | n | β | 95% CI | β | 95% CI | β | 95% CI | β | 95% CI |
| Total[e] | 186 | 1.6 | 0.9, 2.3 | 2.0 | 1.4, 2.6 | 2.1 | 1.5, 2.6 | 1.9 | 1.4, 2.5 |
| E-COPD | 101 | 1.6 | 0.7, 2.4 | 2.0 | 1.3, 2.7 | 2.2 | 1.5, 2.9 | 2.2 | 1.4, 2.9 |
| NE-COPD | 85 | 1.6 | 0.3, 2.9 | 2.0 | 0.9, 3.1 | 1.6 | 0.6, 2.6 | 1.2 | 0.2, 2.2 |

$FEV_{1Z-score}$–forced expiratory lung volum in one second Z-score, $VO_{2peak}$–peak oxygen uptake, COPD–chronic obstructive pulmonary disease, β–regression coefficient, CI–confidence interval.

[a]Crude association.

[b]Adjusted for sex and age (years).

[c]Adjusted for sex, age, body mass index (BMI) and smoking status (former, current).

[d]Adjusted for sex, age, BMI, smoking status, systemic beta-blocker (yes, no) and inhaled bronchodilator (yes, no).

[e]Adjusted for COPD phenotype in all total sample models.

Stratified by COPD phenotype, one unit reduction in $FEV_{1Z\text{-score}}$ was, on average, associated with 2.2 (95% CI 1.4, 2.9) and 1.2 (95% CI 0.2, 2.2) ml/kg/min lower $VO_{2peak}$ in E-COPD and NE-COPD, respectively after adjustment for sex, age, BMI, smoking status, beta-blocker and bronchodilator treatment (Table 3). The association of $FEV_{1Z\text{-score}}$ with $VO_{2peak}$ did not differ statistically by phenotype (p-value for interaction = 0.153).

## Associations of standardized $ppFEV_1$ with $VO_{2peak}$ and effect modification by COPD phenotype

Adjusted for COPD phenotype, sex, age, BMI, smoking status, beta-blocker and bronchodilator treatment, one unit reduction in standardized $ppFEV_1$ (i.e. one SD reduction in $ppFEV_1$) was, on average, associated with 2.5 (95% CI 1.8, 3.2) ml/kg/min lower $VO_{2peak}$ (Table 4).

Stratified by COPD phenotype, one unit reduction in standardized $ppFEV_1$ was, on average, associated with 2.7 (95% CI 1.8, 3.6) and 1.6 (95% CI 0.3, 2.9) ml/kg/min lower $VO_{2peak}$ in E-COPD and NE-COPD, respectively after adjustment for sex, age, BMI, smoking status, beta-blocker and bronchodilator treatment (Table 4). The association of standardized $ppFEV_1$ with $VO_{2peak}$ did not differ statistically by phenotype (p-value for interaction = 0.220).

## Phenotype related differences in $VO_{2peak}$ between $FEV_{1Z\text{-score}}$ and $ppFEV_1$

Adjusted for sex, age, BMI, smoking status, beta-blocker and bronchodilator treatment, $VO_{2peak}$ was, on average, 2.2 (95% CI 0.8, 3.6) and 2.1 (95% CI 0.8, 3.5) ml/kg/min lower in E-COPD than in NE-COPD, when comparisons were made at similar $FEV_{1Z\text{-score}}$ and $ppFEV_1$, respectively.

## Sensitivity analyses

In the sensitivity analyses including only those with reduced VR and/or HRR, one unit reduction in $FEV_{1Z\text{-score}}$ was, on average, associated with 2.2 (95% CI 1.5, 2.8) ml/kg/min lower $VO_{2peak}$ after adjustment for COPD phenotype, sex, age, BMI, smoking status, beta-blocker and bronchodilator treatment.

Stratified by COPD phenotype, one unit reduction in $FEV_{1Z\text{-score}}$ was, on average, associated with 2.5 (95% CI 1.6, 3.4) and 1.3 (95% CI 0.3, 2.3) ml/kg/min lower $VO_{2peak}$ in E-COPD (n = 68) and NE-COPD (n = 66), respectively after adjustment for sex, age, BMI, smoking

**Table 4. Associations of standardized $ppFEV_1$ with $VO_{2peak}$ (ml/kg/min) in total cohort and stratified by emphysema (E) and non-emphysema (NE) COPD phenotypes.**

|  |  | Model 1[a] | | Model 2[b] | | Model 3[c] | | Model 4[d] | |
|---|---|---|---|---|---|---|---|---|---|
|  | n | β | 95% CI | β | 95% CI | β | 95% CI | β | 95% CI |
| Total[e] | 186 | 2.5 | 1.6, 3.3 | 2.6 | 1.8, 3.3 | 2.7 | 2.0, 3.3 | 2.5 | 1.8, 3.2 |
| E-COPD | 101 | 2.4 | 1.4, 3.3 | 2.5 | 1.7, 3.3 | 2.7 | 1.9, 3.5 | 2.7 | 1.8, 3.6 |
| NE-COPD | 85 | 2.6 | 0.9, 4.3 | 2.6 | 1.2, 4.0 | 2.0 | 0.7, 3.3 | 1.6 | 0.3, 2.9 |

$ppFEV_1$ –percent predicted forced expiratory lung volum in one second, $VO_{2peak}$ –peak oxygen uptake, COPD–chronic obstructive pulmonary disease, β–regression coefficient, CI–confidence interval.

[a]Crude association.

[b]Adjusted for sex and age (years).

[c]Adjusted for sex, age, body mass index (BMI) and smoking status (former, current).

[d]Adjusted for sex, age, BMI, smoking status, systemic beta-blocker (yes, no) and inhaled bronchodilator (yes, no).

[e]Adjusted for COPD phenotype in all total sample models.

status, beta-blocker and bronchodilator treatment. The association of $FEV_{1Z\text{-}score}$ with $VO_{2peak}$ did not differ statistically by phenotype (p-value for interaction = 0.051).

Adjusted for sex, age, BMI, smoking status, beta-blocker and bronchodilator treatment, $VO_{2peak}$ was, on average, 2.0 (95% CI 0.3, 3.6) ml/kg/min lower in E-COPD than in NE-COPD, when comparisons were made at similar $FEV_{1Z\text{-}score}$.

## Discussion

In this study we found $FEV_{1Z\text{-}score}$ to be positively associated with $VO_{2peak}$. This association was stronger in E-COPD than in NE-COPD but did not differ statistically by phenotype. In corresponding analyses with standardized $ppFEV_1$, similar estimates were observed. Exercise capacity was lower in E-COPD than in NE-COPD, but the phenotype related differences in $VO_{2peak}$ were similar when comparisons were made using $FEV_{1Z\text{-}score}$ and $ppFEV_1$.

Previous studies have reported weak positive associations between measures of dynamic lung volumes and exercise capacity in patients with COPD [8, 9]. Weak positive correlations between $FEV_1$ and $VO_{2peak}$ may be expected in a heterogeneous disease where multiple organ systems and pathophysiological mechanisms interact and contribute in a highly variable manner to exercise limitation in individual patients [23]. However, normalization of $FEV_1$ by using percent predicted may also weaken the association with $VO_{2peak}$, especially in diverse study populations with regards to sex, age and height. The present study is the first to report the association of $FEV_1$ with $VO_{2peak}$ in COPD using Z-scores of $FEV_1/FVC$ and $FEV_1$ to define both the presence of as well as the degree of airway obstruction, respectively. Although, we support the use of Z-scores of dynamic lung volumes in the diagnostic process in COPD, the associations of $FEV_1$ with $VO_{2peak}$ were similar using $FEV_{1Z\text{-}score}$ and standardized $ppFEV_1$. Thus we have no reason to state that Z-scores are preferable over percent predicted regarding the association of $FEV_1$ with $VO_{2peak}$ in COPD. On average, one unit reduction in $FEV_{1Z\text{-}score}$ was associated with 1.9 ml/min/kg (95% CI 1.4, 2.5) lower $VO_{2peak}$. The minimal clinically important difference in $VO_{2peak}$ has not been established in COPD, but we consider this magnitude, representing 9% of the mean $VO_{2peak}$ (20.9 ml/kg/min) in this sample, likely to represent variability in functional impairment.

Several studies have reported differences in exercise capacity for any given reduction in ventilatory capacity when comparing patients with COPD phenotypically stratified by degree of emphysema. O'Donnell et al. [4] reported lower $VO_{2peak}$ in patients with COPD and reduced $T_LCO$ (38 ± 8% predicted) compared to patients with similar $ppFEV_1$ but more preserved $T_LCO$ (73 ± 4% predicted). Farkhooy et al. [24] found $T_LCO$ to be an independent predictor of exercise capacity, assessed by peak workload, in patients with varying severity of COPD. Consistent with the results of these studies, we found lower $VO_{2peak}$ in patients with E-COPD defined by $T_LCO$ below LLN. The estimated difference in $VO_{2peak}$ comparing E-COPD and NE-COPD was 2.2 (95% CI 0.8, 3.6) and 2.1 (95% CI 0.8, 3.5) ml/kg/min when patients were compared at similar $FEV_{1Z\text{-}score}$ and $ppFEV_1$, respectively. Thus, the difference in exercise capacity comparing these two COPD phenotypes seems independent of sex, age and height related variance in $FEV_1$. In fact, lower exercise capacity in E-COPD seems to be conditional on pathophysiological mechanisms unrelated to reduction in ventilatory capacity as estimated from $FEV_1$.

In a small study of patients with COPD and moderate to severe emphysema (n = 16), Paoletti et al. [25] compared the ventilatory response to maximal incremental exercise between patients with degree of emphysema above and below 50% as measured by high resolution computed tomography. In the group with over 50% emphysema, there was evidence of more dynamic hyperinflation, higher tidal volume to inspiratory capacity ratio ($V_T/IC$), higher end-

tidal partial pressure of $CO_2$ ($P_{ET}CO_2$) as well as increasing arterial partial pressure of $CO_2$ ($P_aCO_2$) at peak exercise, suggesting greater mechanical constraints and relative hypoventilation in this group. O'Donnell et al. [4] also found more dynamic hyperinflation at lower exercise levels, more rapid attainment of volume constraints and higher degree of exercise induced dyspnea in patients with COPD and lower compared to higher $T_LCO$. We found both lower VR, suggesting higher ventilatory demand at peak exercise relative to estimated maximal ventilatory capacity, and a higher proportion reporting test termination due to dyspnea in E-COPD compared to NE-COPD. Mechanical ventilatory limitation may therefore be a principal pathophysiological mechanism explaining the difference in exercise capacity between patients with E-COPD and NE-COPD found in our study.

Since lower aerobic capacity in E-COPD seems to be unrelated to ventilatory capacity, we may also hypothesize that greater inclination to mechanical ventilatory limitation is caused by increased ventilatory demand due to more pronounced gas exchange abnormalities and ventilatory inefficiency in E-COPD. Unfortunately, due to technical problems in the initial data extracting process, only peak exercise data were available and measures of ventilatory efficiency were therefore not included in this study. However, Rinaldo et al. [26] recently reported differences in ventilatory efficiency when comparing noninvasive measures between patients with emphysema and non-emphysema COPD phenotypes. The ventilatory equivalents for $CO_2$ (VE/VCO_2), both nadir and slope below ventilatory compensation point, were higher in emphysema indicating more pronounced ventilatory inefficiency in this phenotype.

Although we did not find strong evidence to support an effect modification of COPD phenotype on the association between $FEV_{1Z-score}$ and $VO_{2peak}$ (p-value for interaction 0.153), the estimated association was stronger in E-COPD (2.2 ml/kg/min, 95% CI 1.4, 2.9) than in NE-COPD (1.2 ml/kg/min, 95% CI 0.2, 2.2). Therefore, we cannot exclude the possibility that, additionally to being a marker of ventilatory capacity, any reduction in $FEV_1$ may either impact or reflect gas exchange, ventilation/perfusion (V/Q) heterogeneity and ventilatory efficiency differently depending on COPD phenotype. Interestingly, although not confirmative of such speculation, we did find lower $S_pO_{2min}$ in E-COPD compared to NE-COPD. Future studies are warranted and should aim to address underlying pathophysiological mechanisms explaining differences in exercise capacity comparing E-COPD and NE-COPD. In this regard, we believe invasive measures of gas exchange and calculations of dead space ventilation from arterial blood gasses would be beneficial.

Boutou et al. [27] found percent predicted $T_LCO$ ($ppT_LCO$) to be an independent predictor of survival in patients with COPD, and the impact of $VO_{2peak}$ on mortality is well known in this disease [6]. By confirming a positive association between phenotype assessment by $T_LCO_{Z-score}$ and $VO_{2peak}$, the present study provides additional evidence linking two prognostic measures in COPD. In agreement with previous studies [24–27], we advocate that assessment of $T_LCO$ should be considered in the evaluation of patients with COPD to add clinical information not inferable from simple spirometric measurements alone. Furthermore, the finding of both lower $FEV_{1Z-score}$ and $VO_{2peak}$ in patients with E-COPD, coincide with COPD phenotype being a positive modifier of the association between these key variables. We suggest that differentiation between $T_LCO$ below and above LLN should complement measures of dynamic lung volumes in future studies of exercise capacity in COPD.

A strength of the current study is that we used Z-scores of lung function measures when reporting the association of $FEV_1$ with $VO_{2peak}$. This approach is novel and E-COPD and NE-COPD may be more correctly differentiated by $T_LCO_{Z-score}$, taking sex, age and height related variance into account, than by $ppT_LCO$. Furthermore, confounding effects of sex and age on the association of $FEV_{1z-score}$ with $VO_{2peak}$ are likely to reflect sampling variation, which was explored in a separate model. Additionally, the patients assessed in this study were

referred for evaluation from primary care and may represent patients typically encountered in a specialist clinic for diagnostic or prognostic evaluation of COPD.

This study has limitations. COPD phenotypes were dichotomized into E-COPD and NE-COPD based on $T_LCO$ below and above LLN, respectively. Although the pre-test probability for emphysema is assumed high in this study population consisting of patients with COPD, radiological qualitative and quantitative confirmation by thoracic computed tomography was not available. On the other hand, this is a real life study and assessment of emphysematous component by $T_LCO$ in obstructive lung disease is recommended in clinical practice [14].

The gold standard to assess exercise capacity is direct measurement of $VO_2$ at maximal exercise ($VO_{2max}$). $VO_{2peak}$, used in this study, is susceptible to patient effort and may underestimate true $VO_{2max}$ [28]. Provided that any underestimation of $VO_{2max}$ from $VO_{2peak}$ is unrelated to $FEV_{1Z-score}$, only the precision of the estimates should potentially be influenced. In contrast, associations between underestimation of $VO_{2max}$ from $VO_{2peak}$ and $FEV_{1Z-score}$ could introduce systematic error and bias the results. We identified 52 patients (E-COPD; n = 33) with both VR above 15% and HRR above 10%, potentially indicative of submaximal effort, i.e. both preserved residual capacity of the ventilatory and cardiac system, respectively [22]. Excluding these patients in the sensitivity analyses did not appreciably influence the estimates.

In this cross sectional study we can only test for associations. The nature of the data does not permit causal inference or definite conclusions on involved pathophysiological mechanisms behind observed results. Therefore, the clinical value of differentiation on COPD phenotype, supported by this study, cannot be extended to suggest specific treatment recommendations in individual patients.

## Conclusions

In COPD, $FEV_{1Z-score}$ is positively associated with $VO_{2peak}$. This association was stronger in E-COPD than in NE-COPD but did not differ statistically by phenotype. Both the association of $FEV_1$ with $VO_{2peak}$ and the difference in $VO_{2peak}$ comparing COPD phenotypes seems independent of sex, age and height related variance in $FEV_1$. Mechanisms leading to reduction in $FEV_1$ may contribute to lower $VO_{2peak}$ in E-COPD. Future studies aiming to address underlying pathophysiological mechanisms of these findings are warranted.

## Supporting information

**S1 Dataset. Minimal dataset.**
(SAV)

## Acknowledgments

The authors would like to acknowledge Marina Vang for her contribution as a research assistant in the initial data extraction process.

## Author Contributions

**Conceptualization:** Øystein Rasch-Halvorsen, Erlend Hassel, Ben M. Brumpton, Martijn A. Spruit, Arnulf Langhammer, Sigurd Steinshamn.

**Formal analysis:** Øystein Rasch-Halvorsen.

**Investigation:** Haldor Jenssen.

**Methodology:** Øystein Rasch-Halvorsen, Erlend Hassel, Ben M. Brumpton, Martijn A. Spruit, Arnulf Langhammer, Sigurd Steinshamn.

**Writing – original draft:** Øystein Rasch-Halvorsen, Erlend Hassel, Ben M. Brumpton, Haldor Jenssen, Martijn A. Spruit, Arnulf Langhammer, Sigurd Steinshamn.

**Writing – review & editing:** Øystein Rasch-Halvorsen, Erlend Hassel, Ben M. Brumpton, Haldor Jenssen, Martijn A. Spruit, Arnulf Langhammer, Sigurd Steinshamn.

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
