## [Decision Letter · Decision Letter 0]

25 Jan 2021

PONE-D-20-31837

Lung function and peak oxygen uptake in chronic obstructive pulmonary disease phenotypes with and without emphysema

PLOS ONE

Dear Dr. Øystein Rasch-Halvorsen,

Thank you for submitting your manuscript to PLOS ONE. After careful consideration, we feel that it has merit but does not fully meet PLOS ONE’s publication criteria as it currently stands. Therefore, we invite you to submit a revised version of the manuscript that addresses the points raised during the review process.

We look forward to receiving your revised manuscript.

Kind regards,

Manlio Milanese

Academic Editor

PLOS ONE

Journal Requirements:

2.) Please include the date(s) on which you accessed the databases or records to obtain the data used in your study.

Additional Editor Comments:

This is an intriguing manuscript that can be implemented enough to considered for publication in Plos One. Please follow the suggestions of the reviewer.

This retrospective study evaluating the association between the FEV1-1Zscore and VO2 peak in E and N COPD is intruging and well written, but methodologically too weak and to be suitable for publication in Plos One. The suggestions from the reviewer could implement the manuscript for a second consideration.

Reviewers' comments:

Reviewer's Responses to Questions

**Comments to the Author**

1. Is the manuscript technically sound, and do the data support the conclusions?

Reviewer #1: Partly

2. Has the statistical analysis been performed appropriately and rigorously? 

Reviewer #1: Yes

3. Have the authors made all data underlying the findings in their manuscript fully available?

Reviewer #1: Yes

4. Is the manuscript presented in an intelligible fashion and written in standard English?

Reviewer #1: Yes

5. Review Comments to the Author

Reviewer #1: This is a retrospective study, evaluated the association between the FEV1-1Zscore and VO2 peak in E and N COPD. The authors demonstrated that there is not a statistical difference between the association FEV1-1Zscore- VO2 and the COPD phenotypes.

The idea of this study to associate VO2 with FEV1-1zscore is very interesting and original, and the data are also interesting, but in my opinion the aim of the study is not clear. Authors should compare the associations of FEV1% predicted-VO2 in both groups and compare them to those of FEV1-1zscore-VO2 and see if the association is lost with the predicted ones. Then they would confirm that the weak association of FEV1predicted-VO2 depends on how the predicted ones are calculated, otherwise this work remains just descriptive and becomes just descriptive.

I think the authors should re-analyse the data comparing the correlation of FEV1%pr/VO2 versus FEV1-zscore/VO2.

In table 1 and 2, there are no significant differences, is it correct? Apparently the groups are very different.

6. PLOS authors have the option to publish the peer review history of their article (what does this mean?). If published, this will include your full peer review and any attached files.

Reviewer #1: No

---

## [Author Response · Author response to Decision Letter 0]

15 Mar 2021

Journal Requirements:

Response: 

In the current version of the manuscript, corrections have been made in order to meet PLOS ONE`s style requirements.

2.) Please include the date(s) on which you accessed the databases or records to obtain the data used in your study.

Response: 

In the revised version of the manuscript, this information is now included. 

Materials and methods, page 6, line 109-110:

“The clinical database was accessed in the time period between the fifteenth of July and the thirtieth of August 2015.”

Additional Editor Comments:

This is an intriguing manuscript that can be implemented enough to considered for publication in Plos One. Please follow the suggestions of the reviewer.

This retrospective study evaluating the association between the FEV1-1Zscore and VO2 peak in E and N COPD is intruging and well written, but methodologically too weak and to be suitable for publication in Plos One. The suggestions from the reviewer could implement the manuscript for a second consideration.

Reviewers' comments:

Reviewer's Responses to Questions

Comments to the Author

1. Is the manuscript technically sound, and do the data support the conclusions?

Reviewer #1: Partly

2. Has the statistical analysis been performed appropriately and rigorously?

Reviewer #1: Yes

3. Have the authors made all data underlying the findings in their manuscript fully available?

Reviewer #1: Yes

4. Is the manuscript presented in an intelligible fashion and written in standard English?

Reviewer #1: Yes

5. Review Comments to the Author

Reviewer #1: This is a retrospective study, evaluated the association between the FEV1-1Zscore and VO2 peak in E and N COPD. The authors demonstrated that there is not a statistical difference between the association FEV1-1Zscore- VO2 and the COPD phenotypes.

The idea of this study to associate VO2 with FEV1-1zscore is very interesting and original, and the data are also interesting, but in my opinion the aim of the study is not clear. 

Response: 

We have rephrased the following sections of the manuscript in order to clarify the aims of the study.

Abstract, page 3, line 31-38:

“Previous studies in chronic obstructive pulmonary disease (COPD) have not taken sex, age and height related variance of dynamic lung volumes into account, when reporting associations of forced expiratory lung volume in one second (FEV1) with peak oxygen uptake (VO2peak) and differences in VO2peak comparing phenotypes characterized by degree emphysema. We aimed to assess the association of FEV1Z-score with VO2peak in COPD (n = 186) and investigate whether this association differs between emphysema (E-COPD) and non-emphysema (NE-COPD) phenotypes. Additionally, phenotype related differences in VO2peak were compared using FEV1Z-score and percent of predicted FEV1 (ppFEV1).

Introduction, page 5-6, line 99-103:

“We aimed to assess the association of FEV1Z-score with VO2peak in COPD and investigate whether this association differs comparing phenotypes with and without emphysema. Additionally, we explored the potential influence of sex, age and height related variance in FEV1 on both the association of FEV1 with VO2peak and on the difference in VO2peak between E-COPD and NE-COPD.”

Reviewer #1: Authors should compare the associations of FEV1% predicted-VO2 in both groups and compare them to those of FEV1-1zscore-VO2 and see if the association is lost with the predicted ones. Then they would confirm that the weak association of FEV1predicted-VO2 depends on how the predicted ones are calculated, otherwise this work remains just descriptive and becomes just descriptive.

I think the authors should re-analyse the data comparing the correlation of FEV1%pr/VO2 versus FEV1-zscore/VO2.

Response:

As suggested by the reviewer we have reanalyzed the data to compare the associations of FEV1Z-score and percent of predicted FEV1 (ppFEV1) with VO2peak.

Given the different units of measurements, we have standardized ppFEV1 in order to compare the regression coefficients with those of FEV1Z-score. 

Additionally, we have extended the analyses to compare the phenotype related difference in VO2peak when adjusting for FEV1Z-score and ppFEV1 in separate models.

The following sections have been revised as follows:

Methods, page 8, line 165-166:

“Corresponding analyses with standardized ppFEV1, calculated as (observed value – sample mean)/SD, were performed for comparison.”

Methods, page 9, line 168-172:

“Linear regression models were also used to investigate whether the difference in VO2peak between E-COPD and NE-COPD is influenced by sex, age and height related variance in FEV1. VO2peak was regressed on COPD phenotype and ppFEV1 and FEV1Z-score in separate total sample models with adjustment for sex, age, BMI, smoking status, beta-blocker and bronchodilator treatment.”

Results, page 14-16, line 252-274:

“Associations of standardized ppFEV1 with VO2peak and effect modification by COPD phenotype. 

Adjusted for COPD phenotype, sex, age, BMI, smoking status, beta-blocker and bronchodilator treatment, one unit reduction in standardized ppFEV1 (i.e. one SD reduction in ppFEV1) was, on average, associated with 2.5 (95% CI 1.8, 3.2) ml/kg/min lower VO2peak (Table 4).

Table 4. Associations of standardized ppFEV1 with VO2peak (ml/kg/min) in total cohort and stratified by emphysema (E) and non-emphysema (NE) COPD phenotypes.

 Model 1a Model 2b Model 3c Model 4d

 n β 95% CI β 95% CI β 95% CI β 95% CI

Totale 186 2.5 1.6, 3.3 2.6 1.8, 3.3 2.7 2.0, 3.3 2.5 1.8, 3.2

E-COPD 101 2.4 1.4, 3.3 2.5 1.7, 3.3 2.7 1.9, 3.5 2.7 1.8, 3.6

NE-COPD 85 2.6 0.9, 4.3 2.6 1.2, 4.0 2.0 0.7, 3.3 1.6 0.3, 2.9

ppFEV1 – percent of predicted forced expiratory lung volum in one second, VO2peak – peak oxygen uptake, COPD – chronic obstructive pulmonary disease, β – regression coefficient, CI – confidence interval. 

aCrude association. bAdjusted for sex and age (years). cAdjusted for sex, age, body mass index (BMI) and smoking status (former, current). dAdjusted for sex, age, BMI, smoking status, systemic beta-blocker (yes, no) and inhaled bronchodilator (yes, no). eAdjusted for COPD phenotype in all total sample models.

Stratified by COPD phenotype, one unit reduction in standardized ppFEV1 was, on average, associated with 2.7 (95% CI 1.8, 3.6) and 1.6 (95% CI 0.3, 2.9) ml/kg/min lower VO2peak in E-COPD and NE-COPD, respectively after adjustment for sex, age, BMI, smoking status, beta-blocker and bronchodilator treatment (Table 4). The association between standardized ppFEV1 and VO2peak did not differ statistically by phenotype (p-value for interaction = 0.220).”

Results, page 16, line 275-280:

“Phenotype related differences in VO2peak between FEV1Z-score and ppFEV1.

Adjusted for sex, age, BMI, smoking status, beta-blocker and bronchodilator treatment, VO2peak was, on average, 2.2 (95% CI 0.8, 3.6) and 2.1 (95% CI 0.8, 3.5) ml/kg/min lower in E-COPD than in NE-COPD, when comparisons were made at similar FEV1Z-score and ppFEV1, respectively.”

Discussion, page 17, line 294-299:

“In this study we found FEV1Z-score to be positively associated with VO2peak. This association was stronger in E-COPD than in NE-COPD but did not differ statistically by phenotype. In corresponding analyses with standardized ppFEV1 similar estimates were observed. The exercise capacity was lower in E-COPD than in NE-COPD, but the phenotype related differences in VO2peak were similar when comparisons were made using FEV1Z-score and ppFEV1.”

Discussion, page 17, line 309-313:

“Although, we support the use of Z-scores of dynamic lung volumes in the diagnostic process in COPD, the associations of FEV1 with VO2peak were similar using FEV1Z-score and standardized ppFEV1. Thus we have no reason to state that Z-scores of FEV1 are preferable over percent of predicted values regarding the association with VO2peak in COPD.”

Discussion, page 18, line 325-329:

“The estimated difference in VO2peak comparing E-COPD and NE-COPD was 2.2 (95% CI 0.8, 3.6) and 2.1 (95% CI 0.8, 3.5) ml/kg/min when patients were compared at similar FEV1Z-score and ppFEV1, respectively. Thus, the difference in exercise capacity comparing these two COPD phenotypes seems independent on whether or not sex, age and height related variance in FEV1 is taken into account.”

Conclusions, page 22, line 413-415:

“Both the association of FEV1 with VO2peak and the difference in VO2peak comparing COPD phenotypes seems independent of sex, age and height related variance in FEV1.”

Abstract, page 3, line 46-49 and line 51-53:

“Similar estimates were obtained in analyses with standardized ppFEV1. Compared to NE-COPD, VO2peak was 2.2 (95% CI 0.8, 3.6) and 2.1 (95% CI 0.8, 3.5) ml/kg/min lower in E-COPD, when adjusted for FEV1Z-score and ppFEV1, respectively.”

“Both the association of FEV1 with VO2peak and the difference in VO2peak comparing COPD phenotypes seems independent of sex, age and height related variance in FEV1.”

Reviewer #1: In table 1 and 2, there are no significant differences, is it correct? Apparently the groups are very different.

Response: 

In the original manuscript several variables presented in table 1 and 2 were tested between groups:

Methods, page 8, line 154-155:

“Lung function and exercise variables were compared between patients with E-COPD and NE-COPD using independent samples t-tests.”

The crude unadjusted difference in the following variables were reported with point estimates and 95% CIs. We have now added p-values in the revised version of the manuscript: 

Results, page 11, line 204-209:

“Compared to patients with NE-COPD, patients with E-COPD had lower FEV1Z-score (difference 0.69, 95% CI 0.36, 1.02, p < 0.001), lower FVCZ-score (difference 0.42, 95% CI 0.05, 0.79, p = 0.028), lower FEV1/FVCZ-score (difference 0.62, 95% CI 0.38, 0.86, p < 0.001), lower KCOZ-score (difference 1.74, 95% CI 1.40, 2.07, p < 0.001), lower VAZ-score (difference 0.91, 95% CI 0.56, 1.27, p < 0.001) and, by definition, lower TLCOZ-score (difference 2.41, 95% CI 2.14, 2.68, p < 0.001) (Table 1).”

Results, page 13, line 226-230:

“Compared to patients with NE-COPD, patients with E-COPD had lower VO2peak (difference 4.8 ml/kg/min, 95% CI 3.0, 6.5, p < 0.001), lower RERpeak (difference 0.04 95% CI 0.004, 0.08, p = 0.031), higher HRR (difference 6.1%, 95% CI 2.5, 9.6, p = 0.001), lower VEpeak (difference 14.8 l/min, 95% CI 7.8, 21.9, p < 0.001), lower VR (difference 5.5%, 95% CI -0.5, 11.5, p = 0.073) and lower SpO2min (difference 1.7%, 95% CI 0.9, 2.6, p < 0.001) (Table 2).”

The implication of these differences are addressed as follows:

Discussion, page 19, line 341-346:

“We found both lower VR, suggesting higher ventilatory demand at peak exercise relative to estimated maximal ventilatory capacity, and a higher proportion reporting test termination due to dyspnea in E-COPD compared to NE-COPD. Mechanical ventilatory limitation may therefore be a principal pathophysiological mechanism explaining the difference in exercise capacity between patients with E-COPD and NE-COPD found in our study.”

Discussion, page 19, line 361-365:

“we cannot exclude the possibility that, additionally to being a marker of ventilatory capacity, any reduction in FEV1 may either impact or reflect gas exchange, ventilaton (V)/perfusion (Q) heterogeneity and ventilatory efficiency differently depending on COPD phenotype. Interestingly, but not confirmative of such speculation, we did find lower SpO2min in E-COPD compared to NE-COPD.”

Discussion, page 20, line 376-380:

“the finding of both lower FEV1Z-score and VO2peak in patients with E-COPD, coincide with COPD phenotype being a positive confounder of the association between these key variables. We suggest that differentiation between TLCO below and above LLN should complement measures of dynamic lung volumes in future studies of exer

---

## [Decision Letter · Decision Letter 1]

21 Apr 2021

PONE-D-20-31837R1

Lung function and peak oxygen uptake in chronic obstructive pulmonary disease phenotypes with and without emphysema

PLOS ONE

Dear Dr. Øystein Rasch-Halvorsen,

Thank you for submitting your manuscript to PLOS ONE. After careful consideration, we feel that it has merit but does not fully meet PLOS ONE’s publication criteria as it currently stands. Therefore, we invite you to submit a revised version of the manuscript that addresses the points raised during the review process.

minor revision. Check English language and its fluency

We look forward to receiving your revised manuscript.

Kind regards,

Manlio Milanese

Academic Editor

PLOS ONE

Journal Requirements:

Additional Editor Comments (if provided):

Please, find the final comments of the reviewer and submit the manusucript to an English writer to improve the quality of the language

Reviewers' comments:

Reviewer's Responses to Questions

**Comments to the Author**

1. If the authors have adequately addressed your comments raised in a previous round of review and you feel that this manuscript is now acceptable for publication, you may indicate that here to bypass the “Comments to the Author” section, enter your conflict of interest statement in the “Confidential to Editor” section, and submit your "Accept" recommendation.

Reviewer #1: All comments have been addressed

2. Is the manuscript technically sound, and do the data support the conclusions?

Reviewer #1: Yes

3. Has the statistical analysis been performed appropriately and rigorously? 

Reviewer #1: Yes

4. Have the authors made all data underlying the findings in their manuscript fully available?

Reviewer #1: Yes

5. Is the manuscript presented in an intelligible fashion and written in standard English?

Reviewer #1: Yes

6. Review Comments to the Author

Reviewer #1: The authors modified the paper according to the indications.

They should put the significances on the tables (table 1 and 2), because the two populations look different * < 0.05 or ** < 0.01

Line 309 is in BOLD, please change it.

7. PLOS authors have the option to publish the peer review history of their article (what does this mean?). If published, this will include your full peer review and any attached files.

Reviewer #1: No

---

## [Author Response · Author response to Decision Letter 1]

28 Apr 2021

Journal Requirements:

Response:

The reference list has been reviewed and found to be complete and correct. No retracted papers have been cited. There are no changes to the reference list.

Additional Editor Comments (if provided):

Please, find the final comments of the reviewer and submit the manuscript to an English writer to improve the quality of the language

Response:

With specific intention to improve the quality of the language as requested, the current version of the manuscript has been reviewed and revised by Ben M. Brumpton who is a native speaker of the English language.

Reviewers' comments:

Reviewer's Responses to Questions

Comments to the Author

1. If the authors have adequately addressed your comments raised in a previous round of review and you feel that this manuscript is now acceptable for publication, you may indicate that here to bypass the “Comments to the Author” section, enter your conflict of interest statement in the “Confidential to Editor” section, and submit your "Accept" recommendation.

Reviewer #1: All comments have been addressed

2. Is the manuscript technically sound, and do the data support the conclusions?

Reviewer #1: Yes

3. Has the statistical analysis been performed appropriately and rigorously?

Reviewer #1: Yes

4. Have the authors made all data underlying the findings in their manuscript fully available?

Reviewer #1: Yes

5. Is the manuscript presented in an intelligible fashion and written in standard English?

Reviewer #1: Yes

6. Review Comments to the Author

Reviewer #1: The authors modified the paper according to the indications.

They should put the significances on the tables (table 1 and 2), because the two 

populations look different * < 0.05 or ** < 0.01

Response:

As requested by the reviewer, the p-values comparing selected continuous variables between E-COPD and NE-COPD have been included in tables 1 and 2.

Line 309 is in BOLD, please change it.

Response:

Corrected.

---

## [Decision Letter · Decision Letter 2]

17 May 2021

Lung function and peak oxygen uptake in chronic obstructive pulmonary disease phenotypes with and without emphysema

PONE-D-20-31837R2

Dear Dr. Øystein Rasch-Halvorsen,

We’re pleased to inform you that your manuscript has been judged scientifically suitable for publication and will be formally accepted for publication once it meets all outstanding technical requirements.

Kind regards,

Manlio Milanese

Academic Editor

PLOS ONE

Additional Editor Comments (optional):

The manuscript is now suitable for publication on PlosOne.

Reviewers' comments:

Reviewer's Responses to Questions

**Comments to the Author**

1. If the authors have adequately addressed your comments raised in a previous round of review and you feel that this manuscript is now acceptable for publication, you may indicate that here to bypass the “Comments to the Author” section, enter your conflict of interest statement in the “Confidential to Editor” section, and submit your "Accept" recommendation.

Reviewer #1: All comments have been addressed

2. Is the manuscript technically sound, and do the data support the conclusions?

Reviewer #1: Yes

3. Has the statistical analysis been performed appropriately and rigorously? 

Reviewer #1: Yes

4. Have the authors made all data underlying the findings in their manuscript fully available?

Reviewer #1: Yes

5. Is the manuscript presented in an intelligible fashion and written in standard English?

Reviewer #1: Yes

6. Review Comments to the Author

Reviewer #1: the authors answered reviewers' questions by improving the quality of the paper. It would be advisable to put all the p's in tables 1 and 2 and not just the significant ones.

7. PLOS authors have the option to publish the peer review history of their article (what does this mean?). If published, this will include your full peer review and any attached files.

Reviewer #1: No

---

## [Editor Report · Acceptance letter]

19 May 2021

PONE-D-20-31837R2 

Lung function and peak oxygen uptake in chronic obstructive pulmonary disease phenotypes with and without emphysema 

Dear Dr. Rasch-Halvorsen:

I'm pleased to inform you that your manuscript has been deemed suitable for publication in PLOS ONE. Congratulations! Your manuscript is now with our production department. 

Kind regards, 

on behalf of

Dr. Manlio Milanese 

Academic Editor

PLOS ONE